# Probiotics in the Therapeutic Arsenal of Dermatologists

**DOI:** 10.3390/microorganisms9071513

**Published:** 2021-07-15

**Authors:** Vicente Navarro-López, Eva Núñez-Delegido, Beatriz Ruzafa-Costas, Pedro Sánchez-Pellicer, Juan Agüera-Santos, Laura Navarro-Moratalla

**Affiliations:** 1MiBioPath Research Group, Health and Science Faculty, Catholic University of Murcia, Campus de los Jerónimos n 135, 30107 Murcia, Spain; eva.nunez@bioithas.com (E.N.-D.); beatriz.ruzafa@bioithas.com (B.R.-C.); pedro.sanchez@bioithas.com (P.S.-P.); juan.aguera@bioithas.com (J.A.-S.); laura.navarro@bioithas.com (L.N.-M.); 2Clinical Microbiology and Infectious Disease Unit, Hospital Universitario Vinalopó, 03293 Elche, Spain

**Keywords:** probiotic, skin microbiome, gut dysbiosis, atopic dermatitis, acne, psoriasis, microbiota

## Abstract

During the last years, numerous studies have described the presence of significant gut and skin dysbiosis in some dermatological diseases such as atopic dermatitis, psoriasis and acne, among others. How the skin and the gut microbiome play a role in those skin conditions is something to explore, which will shed light on understanding the origin and implication of the microbiota in their pathophysiology. Several studies provide evidence for the influence of probiotic treatments that target the modulation of the skin and intestinal microbiota in those disorders and a positive influence of orally administered probiotics on the course of these dermatosis. The pathologies in which the therapeutic role of the probiotic has been explored are mainly atopic dermatitis, psoriasis and acne. This article aims to review these three dermatological diseases, their relationship with the human microbiota and specially the effect of probiotics usage. In addition, the pathophysiology in each of them and the hypotheses about possible mechanisms of the action of probiotics will be described.

## 1. Introduction

Probiotics are defined as those microorganisms that, when administered in sufficient quantities, confer a health benefit [1]. Some pathologies in which dysbiosis is present and the therapeutic role of probiotics has been explored are atopic dermatitis, psoriasis and acne [2]. The pathophysiology in each of them, the hypotheses about the role of gut and skin dysbiosis and possible mechanisms of action of some different strains of probiotics will be reviewed in this article. To find the information and references included in this systematic review, all authors searched electronic literature databases (mainly https://pubmed.ncbi.nlm.nih.gov and https://europepmc.org/, accessed on 4 July 2021) and proposed a total of 210 articles. All these articles were revised by one author, who contacted some experts for more information on the topic and finally made the decision about the references to be included in this review.

### 1.1. The Microbiota of the Skin

The skin is the organ with the largest surface area in the human body. It serves to separate and protect us from the environment and one of its main functions is to serve as a physical barrier against external agents. Its ecosystem is made up of diverse habitats that harbor a large number of saprophytic microorganisms, including bacteria, fungi, and viruses, as well as some mites. Many of them are harmless or may even perform beneficial functions for the individual. For example, they help protect us against the invasion of pathogenic organisms through their settlement in different epithelial niches and also have an important role in the maduration of skin T cells [3,4].

The skin microbiota is made up of four main bacterial phyla: Actinobacteria, Firmicutes, Proteobacteria, and Bacteroidetes, and of more than 40 identified bacterial genera. Depending on the body area and the individual themself, the proportions of these vary. In sebaceous areas, the genus *Propionibacterium* predominates, while *Staphylococcus* and *Corynebacterium* are more abundant in areas with moist skin. Gram-negative bacterial genera represent the majority in dry skin [5].

Some pathological conditions or factors such as age, diet, or antibiotic consumption, may affect to the normal microbial composition of the individual, that are known as dysbiosis. An unbalanced microbiota can lead to the activation of the immune system and the compromise of the protective function of the epithelial barrier, resulting in the establishment of a pro-inflammatory microbial community and in a clinical condition of chronic inflammation. There are increasing studies that demonstrate the relationship between some pathological conditions of the skin, such as the atopic dermatitis, psoriasis, acne, or rosacea and microbial ecological dysbiosis [6].

### 1.2. The Intestinal Microbiota and Its Involvement in Dermatological Processes

Thanks to High Throughput Sequencing technology, today we have extensive knowledge about the microorganisms (mainly bacteria, but also fungi, viruses, and protozoa) that colonize our intestines and that make up the so-called intestinal microbiota, as well as many of its functions. The scientific evidence that we have today attributes numerous health benefits to the bacteria that are part of our microbiota. Among them, it should be noted its contribution to the degradation of complex indigestible polysaccharides and its essential role in the production of certain nutrients such as vitamin K; It also influences the regulation of the immune response through its ability to differentiate dietary and environmental antigens, protecting the body against pathogens. The intestinal commensal bacteria also intervene in the adaptive immune system by inducing the passage of secretory IgA (sIgA) from the intestinal barrier to IgA, an anti-inflammatory antibody specialized in the protection of the intestinal mucosa against attack by microorganisms. On the other hand, short chain fatty acids (SCFAs) from the fermentation of dietary fiber by bacteria in the gastrointestinal tract play a protective role against the appearance of certain inflammatory disorders, such as arthritis, allergies, and colitis [7].

But what is the role of the gut microbiota in cutaneous homeostasis? Several studies document the immunological and metabolic impact of the intestinal microbiota on other organs of the body, including the skin, through the mechanisms of action of commensal bacteria and their metabolites [8]. If an intestinal dysbiosis occurs, that is, a loss of balance in the individual’s habitual microbial composition, the intestinal barrier may be affected so that it increases its permeability and, thus, a bacterial and intestinal metabolite translocation into the bloodstream is possible [9]. This fact has been observed in patients with psoriasis, in whom intestinal bacterial DNA has been isolated in blood samples when they present disease activity [10]. The SCFAs propionate, acetate, and butyrate, coming from the intestinal fermentation of dietary fiber, are decisive in the fact that the phenomenon of bacterial translocation appears. Those patients who have an intestinal microbiome rich in bacteria that produce these SCFAs have a lower tendency to suffer bacterial translocation phenomena. This phenomenon may be partly responsible for the interconnection between the intestinal and skin microbiota, conditioning the composition of the skin’s own microbiota, as this DNA and bacterial metabolites of intestinal origin present in the blood act on keratinocytes and skin T cells. Ultimately, this activation would provoke an immune and metabolic response of the skin, which would affect the microbial composition of this organ itself [9,11]. The connection between gut and skin microbiota is represented in Figure 1.

## 2. Atopic Dermatitis

Atopic dermatitis is the most common chronic inflammatory disease of the skin, characterized by itching with exudate, xerosis, eczema, and a course marked by flare-ups [12]. Its onset usually occurs at an early age. In around 50% of patients, it begins in the first year of life, while only part continues manifesting in adulthood. Its prevalence has increased considerably in recent years and nowadays it is between 2 and 10% in adults, ranging from one region to another, and from 15 to 30% in children [13,14]. It is strongly associated and can coexist with other allergic, immunological or food intolerances. [15]

The etiopathogenesis of the disease is of multifactorial origin and there are many causes that could trigger it. Among them, we could highlight the mutation in filaggrin, a membrane protein, which compromises the state of the skin’s barrier, causing water loss, increasing the xerosis of the individual and allowing entry and contact with allergens and irritants [16]. From the immunological point of view, an imbalance occurs between Th1 and Th2 cells in favor of the latter, increasing the production of proinflammatory interleukins and immunoglobulin E and thus causing an inflammatory process [17].

The representative lesions of atopic dermatitis are diverse and range from milder forms, such as xerosis, eczema, pityriasis alba or follicular keratosis; even severe forms, such as erythrodermic rash. The diagnosis of atopic dermatitis is clinical and is usually made using the Hanifin and Rajka criteria, which include these typical manifestations, as well as family history and other data from the personal and clinical history of the patients [18].

On the other hand, the characteristic manifestations also vary according to the age range, being observed in different areas in each of the different stages. Three major stages with typically distributed lesions stand out [19]: (1) Infants or children up to two months: predominance of the head and extensor faces of the extremities; (2) Children up to puberty: predominance of skin folds and the backs of the hands and feet and (3) Adults with a combination of both, highlighting folds and extensor surface.

The most widespread and validated main variable for the assessment of atopic dermatitis lesions is the Scoring Atopic Dermatitis (SCORAD) index. It consists of an indicator that scores the extent of the injuries by the total body surface area, assigning an indicative percentage to each area of the body. On the other hand, it assesses the intensity of five fundamental lesions (0 to 3 points): erythema, edema, exudate, excoriation and lichenification, in addition to dryness in the non-compromised areas. Finally, it evaluates the subjective symptoms caused by these lesions, pruritus and loss of sleep, which are assessed on a visual analogical scale (0 to 10 points), a score provided by the parents of the patients, in the case of young children patients, or by the patient themself when they are adults or adolescents [20]. In addition to this scale, we find multiple other scales such as the Eczema Area and Severity Index (EASI) or Investigators’ Global Assessment (IGA), including those that assess the quality of life of both patients and their families when dealing with minors [21,22].

The control of atopic dermatitis, although apparently simple, causes high costs to the health system and problems in the family unit. The usual treatment consists of the use of topical corticosteroids to control the lesions and H1 antihistamines to control the pruritus. Other treatments also used are calcineurin inhibitors, oral corticosteroids in the most serious cases, topical antibiotics to treat infected lesions, phototherapy and biological treatments and monoclonal antibodies for conventional treatment failures. In general, treatments have side effects and are often not effective in completely controlling the symptoms of atopic dermatitis [23,24].

### 2.1. The Skin Microbiota in Atopic Dermatitis

The microbiota of the skin contributes to the maintenance of the balance and homeostasis of the skin, controlling factors such as humidity, temperature, or pH. In general, in the skin microbiota there are several genera that mainly standout such as: *Corynebacterium*, *Propionibacterium* and *Staphylococcus* [25].

The dysbiosis that occurs in the skin’s microbiota in certain situations, promotes the development of diseases, including atopic dermatitis, psoriasis, or acne, among others. In the specific case of atopic dermatitis, *Staphylococcus aureus* is detected and isolated in skin samples affected by this pathology, ranging in a percentage of between 30 and 100% of patients. If we focus on active lesions, it is detected in 70% of them. Therefore, it can be stated that its presence is an important part of atopy, although it remains to be clarified whether these changes are at the origin or are a consequence of the disease itself [26]. What does seem clear is the relationship between this species and the prognosis and the evolution of the disease. An increase in *Staphylococcus aureus* is related to the severity of atopic dermatitis and the clinical manifestations seen in outbreaks [27].

### 2.2. The Gut Microbiota in Atopic Dermatitis

The intestinal microbiota of patients with atopic dermatitis is characteristic of these patients and different from that of the general population, so the relationship between atopic dermatitis and intestinal microbiota seems clear. Most studies point to a greater biodiversity present in patients with atopic dermatitis when compared with a healthy population [28].

Regarding genera or specific species, an increase in *Faecalibacterium prausnitzii* has been detected, a fact that behaves contrary to that found in other pathologies in which the intestinal microbiota seems to play a pathophysiological role. This data could be explained by the coexistence of minor intraspecies varieties. It is known that *Faecalibacterium prausnitzii* is a bacterium that induces the production of butyrate, but in the case of atopic dermatitis it is hypothesized that non-producing subspecies are those that predominate in the intestinal tract and that due to this fact they would not provide benefits to the general situation of the patient [29]. Furthermore, *Faecalibacterium* genus showed the highest presence and significant positive correlation with AD severity (SCORAD index) [30].

### 2.3. Probiotics in Atopic Dermatitis

Atopic dermatitis is the skin disease where the majority of studies carried out to evaluate the effect of probiotics on the evolution of the disease are focused. In the study published by our group, Navarro et al., a probiotic composed of *Bifidobacterium lactis, Bifidobacterium longum* and *Lactobacillus casei* administered for 12 weeks was tested, observing clinical and statistically significant differences at the end of the intervention period, achieving a reduction in the SCORAD index of 80% in the probiotic group [31]. In gut microbiota, genera *Bacteroides, Ruminococcus*, and *Bifidobacterium* significantly increased their levels while *Faecalibacterium* decreased after probiotic consumption. [30]

Other relevant studies on these pathologies are collected in various meta-analyses, the most relevant being those published by Chang et al. [32] and Tan-Lim et al. [33]. In summary, the conclusions of some of these studies are detailed below and summarized in Table 1.

The study carried out by Passeron et al. did not show significant differences in the SCORAD index after 12 weeks of treatment with *Lactobacillus rhamnosus*. Both groups reduced SCORAD in a similar way and there were no differences in secondary variables either [34].

Gerasimov et al. describe a significantly higher decrease in SCORAD in the intervention group after 8 weeks of treatment with two specific strains of *Lactobacillus acidophilus* and *Bifidobacterium lactis* together with prebiotics [35].

The study carried out by van der Aa et al. It does not describe differences in SCORAD, but it does confirm the presence of *Bifidobacterium breve* in stools after 12 weeks of ingestion, a bacterium that was not detected in the stool of the patients at the beginning of the study [36].

Shafiei et al. conducted an 8-week study with a treatment composed of seven probiotic strains in which they did not obtain significant differences compared to placebo [37].

In the study developed by Farid et al., similar to the one described above, a mixture of probiotics and prebiotics did obtain differences between groups in favor of probiotic treatment in the evolution of SCORAD after 8 weeks of treatment [38].

Wu et al. detected a difference of more than 50% between groups in favor of probiotic treatment in the SCORAD index, which represents a significant reduction after 10 weeks of ingestion of *Lactobacillus salivarius* together with fructooligosaccharides [39].

To conclude and as a summary of all these works, the probiotic treatment that according to the bibliography would have greater efficacy in atopic dermatitis, should include strains of *Lactobacillus* and *Bifidobacterium* (a combination of several strains) and the treatment should be extended at least to 12 weeks. The benefit appears greater in patients who are more than three years old, and in those with a family history of the disease.

## 3. Psoriasis

Psoriasis is a systemic inflammatory disease characterized by scaly lesioned plaques with defined borders. These lesions are mainly located on the scalp and large areas of the extremities but can occur at any site of the body. The prevalence of psoriasis is around 1–3%, with differences between countries, corresponding the highest prevalence to Western countries. It causes high costs to the health system, as well as a strong psychological impact on patients who suffer from it. The diagnosis of psoriasis is clinical, there is no specific laboratory parameter of the disease and in most cases, it is not necessary to perform a histological confirmation [40,41].

The etiopathogenesis of psoriasis is not fully known, although most authors postulate that it would be a skin disorder of genetic origin, finally triggered by external factors, that would cause changes at the immune level. The disease is associated with inflammation in other systems and organs, as evidenced by the fact of finding a correlation with inflammatory bowel disease, where between 7 and 11% of diagnosed patients also suffer from psoriasis [4]. Other components as triggers of psoriasis are age, the comorbidity, environmental and external factors [42].

The different variants of psoriasis that are distinguished, are classified according to their symptoms and the characteristics of their lesions and location: (1) Vulgaris or plaque psoriasis, corresponding to 90% of cases; (2) Inverse or inverted psoriasis, also called flexure psoriasis; (3) Guttate psoriasis with children and adolescents being more greatly affected; (4) Pustular psoriasis that presents pustules with a rapid progression; (5) Erythrodermic psoriasis, the most severe type of psoriasis.

Patients with psoriasis usually present other manifestations such as psoriatic arthritis, nail involvement, increased risk of type 2 diabetes, hypertension, hyperlipidemia and coronary heart disease [43]. The clinical activity of the disease is routinely assessed using the Psoriasis Area Surface Index (PASI) scale. The PASI assesses the location of the lesions, as well as their appearance and severity. Erythema, infiltration, and scaling of lesions on the head, trunk, and upper and lower extremities are evaluated (from 0 to 4 points). A 75% reduction in PASI is considered successful treatment and a good prognosis of the disease, improving the quality of life of patients. Achieving PASI 75 is one of the main goals of psoriasis treatment [44].

At first, psoriasis was considered a hyperproliferative disorder, so the objective of its treatment was to reduce this proliferation. Treatment was based on immunosuppressants, either topical or systemic, to try to slow down the immune component of the disease and thereby attenuate the symptoms, but since the eighties in the last century, action began on the production of cytokines as a therapeutic target. Finally, and with the findings in the immunological field, treatments have focused on preventing and controlling production of interleukins such as IL-12, IL-17, IL-20 or IL-23. In these cases, drugs have been developed immune-modulators as anti -TNF and anti-IL-23 monoclonal antibodies, among others [45]. Although these immunomodulatory treatments are more effective, they are expensive and can cause significant adverse effects, which is why they are reserved for the most serious cases. On the other hand, tolerance to them can occur through the production of antibodies, which are no longer effective. In addition to those described treatments, in certain cases, treatment is performed using UVB or PUVA phototherapy [46].

### 3.1. The Skin Microbiota in Psoriasis

The role of the skin microbiota in psoriasis has been little studied, and we have little data on whether it plays any relevant role in the origin of the disease. Some authors defend that, in the case of psoriasis, the skin microbiota does not play a fundamental role and that the progression of the disease is not linked to its composition [47]. However, other authors postulate of a decrease in *Cutibacterium acnes* as a possible cause of the disease and its outbreaks [48]. In other research, significantly lower levels of the phylum Firmicutes and genus *Staphylococcus* were detected in skin that presented lesions, as opposed to healthy skin [49]. In any case, more studies are needed in this regard and above all to assess whether the changes observed so far are at the origin or whether they are simply a consequence of the disease.

### 3.2. The Gut Microbiota in Psoriasis

The intestinal microbiota of patients with psoriasis, which is characteristic and typical of them, presents clear differences with the intestinal microbiota of healthy patients. In several studies it is confirmed that there is a dysbiosis in the intestinal microbiota of patients with psoriasis [50,51,52]. Despite this and in a similar way to what happens with the skin microbiota, there is no defined consensus in these works and research continues to finish describing the composition of the intestinal microbiota responsible for the origin of disease in these patients. Some studies describe a decrease in beneficial genera in the microbiota such as *Parabacteroides* or *Coprobacillus*, like that which occurs in inflammatory bowel disease, a pathology that is correlated with psoriasis as previously indicated. These low levels can ultimately end in a bad immune regulation [53]. Drago et al. proposed a decrease in *Propionibacterium*, and Actinobacteria, while Firmicutes, Proteobacteria, Acidobacteria, *Schlegelella,* Streptococcaeae, Rhodobacteracaea, Campylobacteraceae and Moraxcellaceae would be increased [51]. In Benhadou’s study, *Corynebacterium*, *Propionibacterium*, *Staphylococcus* and *Streptococcus* have been identified as major genera in psoriasis patients [52].

Regarding the correlation between microbiota and the presence of skin lesions, it is described the presence of Firmicutes as being the most abundant in patients with compromised skin while Actinobacteria was decreased in those situations, compared to the microbiota of people with healthy skin or those with psoriasis but without cutaneous activity [52]. In the study published by our research group in 2018 (Codoñer et al.) an increase in *Faecalibacterium* and a decrease in *Bacteroides* are described. In addition to this, an increase in the *Akkermansia* and *Ruminococcus* genera is found in this study [53]. Finally, it is worth highlighting the description of the presence of bacterial DNA in peripheral blood in patients with psoriasis described by our group work in 2015, a situation already perceived in other inflammatory bowel diseases such as inflammatory bowel disease. This phenomenon, as has been indicated elsewhere, could be at the origin of the immunological activation at the intestinal level and secondarily, the presence of DNA and circulating bacterial lipopolysaccharides, be the cause of the activation of keratinocytes, lymphocytes, and other skin cells phenomenon, present in this disease [10]. All these findings appear in Table 2.

### 3.3. Probiotics in Psoriasis

At present, there are no multiple clinical trials where probiotics have been tested as a treatment for psoriasis, so it is a field that has yet to be investigated and contrasted. In the clinical trial carried out by our research team (Navarro-López et al.), in a group of 80 patients with plaque psoriasis, significant differences were observed in disease progression when comparing the subgroup of cases which was administered a probiotic mixture of 12 weeks, compared to those taking placebo. The probiotic mixture used in this study was composed of *Bifidobacterium longum*, *Bifidobacterium lactis* and *Lactobacillus rhamnosus*. At the end of the study, patients from the probiotic group reached PASI75 by 66.7% versus 41.9% by the placebo group [54]. Groeger et al. studied whether supplementation with a strain of *Bifidobacterium infantis* for 6–8 weeks produced changes at the level of cytokines and immunomarkers. In this work, it is concluded that the intake of the probiotic preparation causes a significant reduction in the levels of C-reactive protein and TNF-α when comparing the data with those observed in the placebo group [55]. Finally, a third publication by Vijayshankar et al. which describes the case of a patient with pustular psoriasis to which the probiotic *Lactobacillus sporogenes* together with 10 mg of biotin was administered, in 15 days, a great improvement was observed in the patient, and almost complete bleaching was reached by 4 weeks and remained free of lesions after 6 months of treatment [56].

As a conclusion of the data described on the intestinal microbiota in psoriasis and the little experience with clinical studies using probiotics, we can affirm that the data is very encouraging regarding the adjunctive treatment with probiotics of this disease. The results of our working group are the most conclusive [54], but we will have to wait for more studies with the same probiotic mixture or new preparations that corroborate these first results (Table 3).

## 4. Acne Vulgaris

Acne vulgaris (acne) is a chronic inflammatory skin disease that causes skin lesions and profound negative psychological and social effects on patients reducing their quality of life and productivity [57]. Acne vulgaris is among the most common dermatological conditions worldwide and is the 8th most prevalent disease globally [58]. This skin disease affects to 680 million people with a 10% increase in the last 10 years (data from 2016). Most people experience acne during adolescence, affecting 85% of adolescents and young adults (12–25 years old) in westernized populations [59], and as many as 50% continue to suffer from acne in adulthood [60]. The incidence of acne is higher in females and the severity is greater in males. Interestingly, the incidence of acne vulgaris is greater in wealthy countries [59,60,61].

Skin damage in acne is characterized by non-inflammatory (comedons) and inflammatory (papules, pustules, and nodules) lesions. In clinical research studies, the assessment and grading of acne includes lesion counting, as well as overall grading systems. Overall scales might be less quantitative than lesion counting but more relevant to clinicians and their patients. Currently, no overall acne grading system is considered to be a global standard, although efforts are underway to create a standard [60].

Acne pathophysiology is complex and not completely elucidated. Four core events whose sequence is unknown are involved in the formation of acne lesions: (i) an increment in the sebum produced by the pilosebaceous unit and changes in its composition, (ii) ductal obstruction from increased keratinocyte desquamation and proliferation, (iii) overgrowth of specific strains of *Cutibacterium acnes* (formerly known as *Propionibacterium acnes*), and (iv) systemic and local inflammation. Several factors contribute to acne pathogenesis including androgen imbalance and dysregulation of insulin signaling. Moreover, a high glycaemic Western diet, high consumption of dairy protein, smoking, stress, or modern lifestyle exacerbates acne [57,58,59,60]. Lately, the role of the gut microbiota in dermatological conditions including acne has been pointed out [57,58,59,60,61].

The consensual treatment differentiates the following strategies according to the varying degrees of acne: As considerations prior to the prescription of treatment, in adult women with hormonal predominance acne (it appears mainly on the chin and jaw) the administration of oral contraceptives should be considered. As for the drugs recommended for cases of mild to moderate comedonal acne and papulopustular acne (stages 1 and 2), the local treatment usually suffice. The drug of choice is 2.5%, 5% or 10% POB (benzoyl peroxide-oxidizing and bactericidal agent). Also are useful topical retinoids such as retinoic or adapalene acid. They should be used at a low concentration at the beginning as they have an irritating effect. In stage 3 (severe papulopustular acne and moderate nodular acne), the addition of an oral antibiotic (tetracyclines), along with topical treatment, will be considered. Finally, isotretinoin (retinoid for oral administration) is indicated in the most severe cases, such as those included in stage 4 or as a second choice in the previous stage. These drugs may carry adverse side effects that the professional should consider [62].

### 4.1. Skin Microbiota in Acne

As previously described, the main dominant bacterial phyla on the skin are: Actinobacteria, Proteobacteria, Bacteroidetes, and Firmicutes. More than 60% of the species belong to the genera *Staphylococcus* (Firmicutes), *Corynebacterium* and *Propionibacterium* (Actinobacteria), which vary greatly in their quantity depending on the characteristics of the body region. In areas of oily skin (such as the back, scalp, and face) the lipophilic species of *Propionibacterium* (including *Cutibacterium*) predominate. Moist regions (such as the armpits, the groin area, or the soles of the hands and feet) are rich in *Staphylococcus* and *Corynebacterium* species. Areas of dry skin (such as the forearm) have the most diverse microbial community, with a mixture of all four phylum.

In the pilosebaceous gland, *Cutibacterium acnes* is the principal occupant, it represents up to 90% of the microbiota in the body areas of oily skin. Its role in the pathophysiology of acne has been the subject of study for the last century. However, there is currently no consensus on its involvement in the development of the disease [63]. Other species, such as *Staphylococcus epidermidis*, have been found in more abundant quantities in patients with acne and in patients with active lesions. The participation of *Malassezia* species in acne has also been documented, by observing cases of folliculitis associated with their presence. Thus, some theories suggest that acne could be due more to a microbial interaction than to the mere presence of a specific species [63].

### 4.2. Gut Microbiota in Acne

In a healthy patient, the gut barrier protects from inflammatory molecules. However, during gut microbiota dysbiosis, the gut barrier is compromised, and several molecules can reach the blood system having systemic inflammatory effects including the skin. Recent investigations have suggested the existence of a gut-brain-skin axis where gut microbiota communicates through produced metabolites and other signaling molecules with brain and skin affecting the production of inflammatory molecules in the skin. The connection between acne and gastrointestinal dysfunction can originate in the brain. Supporting this hypothesis is the stress-induced aggravation of acne. In recent years, the role of environmental factors, especially the Western diet, has been raised in acne pathogenesis. Evidence also indicates that the intestinal microbiota associated with the Western diet contribute to inflammatory skin diseases [64].

Few studies have examined the gut microbiota dysbiosis in acne patients observing a decreased diversity, increased Bacteroidetes: Firmicutes ratio (consistent with enterotype of the Western diet), increased Proteobacteria and decreased Actinobacteria phyla, and reduction of genera associated to health for having anti-inflammatory properties such as *Lactobacillus* and *Bifidobacterium* [65]. A research conducted by Deng et al. [65] demonstrated a lower diversity of the gut microbiota in the acne patients included in the study. Also, Yan et al. [66] found remarkable results, with a decrease in *Lactobacillus*, *Bifidobacterium*, *Butyricicoccus*, *Coprobacillus*, and *Allobaculum* in acne patients when their microbiota was compared with that of controls. They would need more studies to identify and characterize these differences to establish the mechanisms linking intestinal dysbiosis and acne.

### 4.3. Probiotics in Acne

The most studied probiotics are those for oral administration due to their direct implication on the intestinal microbiota. However, in the field of Dermatology, there are already some clinical trials that also demonstrate beneficial effects through topical application.

Probiotics given by way topical application, could alter the skin microbiota in a straightforward manner, which would mean a change in some pathological conditions of immune skin response. Among the mechanisms involved would be improving the skin barrier and an increase in secondary production of antimicrobial peptides [67]. Clinical trials in this field are still scarce, but there are already some favorable results for topical probiotic treatment of acne:

The study carried out by Kang et al. used a lotion containing the *Enterococcus faecalis* strain for 8 weeks, resulting in significant improvement in pustule-type acne lesions, compared to the group that used the placebo lotion [68].

A recent clinical trial showed that the administration of *Nitrosomonas eutropha* twice a day for 12 weeks considerably reduced the severity of acne when compared to controls, and a trend towards a reduction in the number of inflammatory lesions was observed [69].

The modulation of the intestinal microbiota through oral probiotic microorganisms can indirectly influence certain dermatological diseases. Gut bacteria have the ability to induce/prevent different pathological states through the gut-skin axis, by which both organs would be communicated through metabolic and neuroendocrine interactions [68]. The most relevant clinical trials conducted with oral probiotics in acne patients are listed here:

In 2013 the team of Jung et al. used a probiotic mixture of *Lactobacillus acidophilus*, *Lactobacillus delbrueckii bulgaricus* and *Bifidobacterium bifidum* in 45 adults with acne. There was a 67% reduction in the lesion count at 12 weeks of treatment, and an 82% reduction when combined with minocycline [70].

Kim et al. showed a decrease of 33.2% in the total count of lesions and 50% in the content of fat at 12 weeks of treatment with *Lactobacillus bulgaricus* and *Streptococcus thermophiles* [71].

In 2016, Fabbrocini et al. administered a *Lactobacillus rhamnosus* strain to a group of 20 individuals with acne for 12 weeks, obtaining a moderate improvement in patients when compared with the placebo group [72].

In conclusion, in the case of acne and likewise with atopic dermatitis and psoriasis, there is data supporting the role of the microbiota in origin and as a result, it is plausible to think that the use of probiotics as adjunctive therapy, will play a relevant role in the future treatment, evolution and prognosis of this disease (Table 4).

## 5. Conclusions

In addition to the pathologies mentioned above, new lines of research are being opened to identify possible benefits of treatment with probiotics in other dermatological disorders. These other pathologies in which the influence of the intestinal microbiota and probiotic treatment on its evolution are currently being studied are rosacea [73], vitiligo [74], seborrheic dermatitis [75], hidradenitis suppurativa [64,75], dandruff [76] and ulcers [77]. The results in these pathologies are very preliminary and there are no clinical studies with a considerable number of patients, so it will be necessary to wait for the results of the works currently in progress to assess the possible efficacy of probiotics in these other skin disorders.

As is clear from the papers reviewed, probiotic therapy to restore an intestinal microbiota and/or skin damaged, may become one important adjuvant therapy in the management of various inflammatory skin diseases, so research in this field bodes well for even greater opportunities for science in different areas of dermatology. There is a lack of more scientific evidence to confirm these data, as well as to better understand the behavior of probiotics in the intestinal microbiota and in the symptoms and manifestations of each pathology. The relationship between skin and microbiota, and its modulation by probiotics, is a field of research with strong scientific interest from which very hopeful and interesting clinical trials can be derived.

## Figures and Tables

**Figure 1 microorganisms-09-01513-f001:**
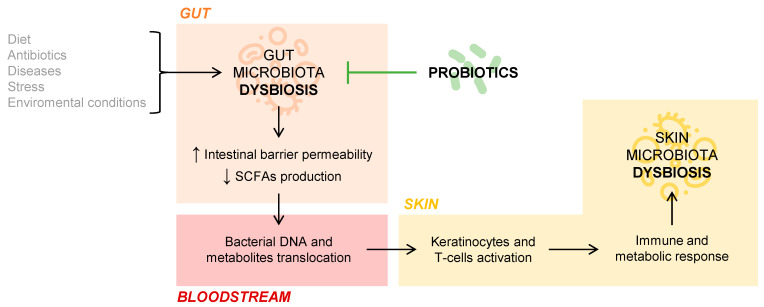
Gut and skin microbiota connection.

**Table 1 microorganisms-09-01513-t001:** Probiotics and atopic dermatitis.

Author	Time of Treatment	Probiotic	Results
Navarro-López et al. [31]	12 weeks	*Bifidobacterium lactis* + *Bifidobacterium longum* + *Lactobacillus casei*	Reduction in SCORAD (PR: −83% vs. PL: −24%; *p* < 0.001) Less use of topical steroids (PR: 7.7% vs. PL: 10.8%)
Climent et al. [30]			Change in gut microbiota composition in PR: Increased in *Bacteroides*, *Ruminococcus*, and *Bifidobacterium* Decreased in *Faecalibacterium*
Passeron et al. [34]	12 weeks	*Lactobacillus rhamnosus rhamnosus* + prebiotics	Not significant differences
Gerasimov et al. [35]	8 weeks	*Lactobacillus acidophilus* + *Bifidobacterium lactis* + prebiotics	Reduction in SCORAD (PR: −33.7% vs. PL: −19.4%; *p* < 0.001) Reduction in IDQOL (PR: −33% vs. PR: −19%; *p* = 0.013) Reduction in DFI (PR: −35.2% vs. PL: −23.8%; *p* = 0.010) Reduction in the use of topical steroids (7.7 g less in PR than in PL; *p* = 0.006)
van der Aa et al. [36]	12 weeks	*Bifidobacterium breve* + prebiotics	No difference in SCORAD between PR and PL Difference in SCORAD in IgE-associated subgroup (PR: −18.1 vs. PL: −13.5, *p* = 0.04) Change in gut microbiota composition: *Bifidobacteria* (PR: 54.7% vs. PL: 30.1%; *p* < 0.001) *Clostridium lituseburense/Clostridium histolyticum* (PR: 0.5 vs. PL: 1.8, *p* = 0.02) *Eubacterium rectale/Clostridium coccoides* (PR: 7.5 vs. PL: 38.1, *p* < 0.001)
Shafiei et al. [37]	8 weeks	A mixture of 7 probiotics strains + prebiotics	Not significant differences
Farid et al. [38]	8 weeks	A mixture of 7 probiotics strains + prebiotics	Reduction in SCORAD (PR: −39.2 vs. PL: −20.10; *p* < 0.005)
Wu et al. [39]	10 weeks	*Lactobacillus salivarius* + prebiotics	Reduction in SCORAD (PR: 27.4 vs. PL: 36.3; *p* < 0.022)

PR: Probiotic group; PL: Placebo group; SCORAD: Scoring Atopic Dermatitis; IDQOL: Infant Dermatitis Quality Of Life; DFI: Dermatitis Family Impact.

**Table 2 microorganisms-09-01513-t002:** Microbiota and psoriasis.

Location	Findings	Reference
Skin microbiota	- Decrease in *Cutibacterium acne*s - Lower levels of Firmicutes and *Staphylococcus*	[47,48,49]
Gut microbiota	- Decrease in *Parabacteroides* or *Coprobacillus* - Decrease in *Propionibacterium* and Actinobacteria; and increase in Firmicutes, Proteobacteria, Acidobacteria, *Schlegelella*, Streptococcaeae, Rhodobacteracaea, Campylobacteraceae and Moraxcellaceae - Major genera: *Corynebacterium*, *Propionibacterium*, *Staphylococcus* and *Streptococcus* - Increase in *Faecalibacterium*, *Akkermansia* and *Ruminococcus* and decrease in *Bacteroides*	[10,52,53]

**Table 3 microorganisms-09-01513-t003:** Probiotics and psoriasis.

Author	Time of Treatment	Probiotic	Results
Navarro-López et al. [54]	12 weeks	*Bifidobacterium longum, Bifidobacterium lactis* and *Lactobacillus rhamnosus*	Reduction in PASI: Patients with PASI reduction up to 75% (PR: 66.7% vs. PL: 41.9%; *p* < 0.05) Follow-up: Lower risk of relapse in PR
Groeger et al. [55]	6–8 weeks	*Bifidobacterium infantis*	Significant reduction in the levels of C-reactive protein and TNF-α in PR
Vijayshankar et al. [56]	4 weeks	*Lactobacillus sporogenes* + biotin	Case report: Complete bleaching

PR: Probiotic group; PL: Placebo group; PASI: Psoriasis Area and Severity Index.

**Table 4 microorganisms-09-01513-t004:** Probiotics and acne.

Author	Time of Treatment	Probiotic	Results
Kang et al. [68]	8 weeks	*Enterococcus faecalis* (topical)	Significant reduction in inflammatory lesions (pustules) in PR
AOBiome [69]	12 weeks	*Nitrosomonas eutropha* (topical)	Reduction in IGA (2-point in PR compared to PL; *p* = 0.03) Higher reduction in number of inflammatory lesions in PR compared to PL
Jung et al. [70]	12 weeks	*Lactobacillus acidophilus*, *Lactobacillus delbrueckii bulgaricus* and *Bifidobacterium bifidum*	Improvement in total lesion count (open label)
Kim et al. [71]	12 weeks	*Lactobacillus bulgaricus* and *Streptococcus thermophiles*	Reduction in inflammatory lesion count (38.6%), total lesion count (23.1%) and acne severity (20.3%) in PR compared to PL Reduction in sebum content (31.1%) in PR compared to PL
Fabbrocini et al. [72]	12 weeks	*Lactobacillus rhamnosus*	Improvement in acne severity (32% in PR; *p* < 0.001)

PR: Probiotic group; PL: Placebo group; IGA: Investigator’s Global Assessment.

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
