# Peer review of "Probiotics in the Therapeutic Arsenal of Dermatologists"

_microorganisms, 2021, doi:10.3390/microorganisms9071513_

Round 1
Reviewer 1 Report
The review paper is written clearly and in detail and the topic is very interesting. Schemes or images that would improve the quality of work are missing. Namely, the tables show the standard methods of treatment while the microbiota is written in the text and it would be good to point out the connection of the intestinal microbiota with the skin microbiota in various chronic diseases. What still catches my eye are the incorrectly written names of the bacteria and the Latin names of the bacteria should be written in Italian. Make sure that the full name is written at the first mention of the bacterium and that the abbreviation is used in the following text. Propionibacterium acne has changed its name and it would be good to indicate this in the text because you use both names.
Author Response
Reviewer 1 Q@A;
1. Schemes or images that would improve the quality of work are missing. Namely, the tables show the standard methods of treatment while the microbiota is written in the text and it would be good to point out the connection of the intestinal microbiota with the skin microbiota in various chronic diseases.
A diagram that shows the relationship between the gut microbiota and the skin microbiota has been added.
The treatment tables have been deleted and summary tables have been added.
2. What still catches my eye are the incorrectly written names of the bacteria and the Latin names of the bacteria should be written in Italian.
Bacterial species and genus names have been corrected and written in Italian.
3. Propionibacterium acne has changed its name and it would be good to indicate this in the text because you use both names.
The name change from Propionibacterium acnes to Cutibacterium acnes has been indicated. Line 346
Reviewer 2 Report
Review for The Gut-Skin Axis: Microbiome, Probiotics and Dermatological Diseases
By Vicente Navarro-Lopez et al.
This review describes the pathophysiology, skin microbiome, gut microbiome, and probiotics in four dermatological diseases.
Major comments:
Reading the abstract suggests already that this review is not renewing from the review “Gut-skin axis : current knowledge of the interrelationship between microbial dysbiosis and skin conditions” that was earlier seen in MICROORGANISMS 9(2). Repetition of four diseases that were also described in there, new information is lacking.
See reference: De Pessermier 2021 Gut–Skin Axis: Current Knowledge of the Interrelationship between Microbial Dysbiosis and Skin Conditions https://www.mdpi.com/2076-2607/9/2/353
And if not mistaken, this paper is not referenced.
Line 92: 1.2. The intestinal microbiota and its involvement in dermatological processes: Only described the impact of the gut microbiome on the skin through the endocrine systems. However, the gut-skin axis is the result of the complex interplay between nervous system, immune system, endocrine systems, and environmental factors, which is worth to mention I believe. In addition, they react with one another in a bidirectional manner.
Minor comments:
Genus taxa are not italicized
Line 31-31 the therapeutic role of the probiotic has been explored include atopic dermatitis with both intervention and primary prevention studies, psoriasis, acne, and rosacea.
The probiotic suggests that there is only one type of probiotic available with therapeutic effects? Suggestion: the therapeutic role of probiotics …
Also, the structure of the sentence needs to be reformed.
Line 70: secretory IgA, sIgA instead of IgA S, IgA, no space between Ig and A
Line 83: The SCFAs instead of The SCFA
Line 99: Reform sentence: it is at between 2 and 10% in adults, ranging from one territory to another, and from 15 to 30% in children
Line 100-101: Add references: It is strongly 100 associated and can coexist with other allergic, immunological or food intolerances.
Line 134-141: Missing references
Table 1: Failure os the above
Line 146: pH
Line 174: Probiotics are defined as those microorganisms that, when administered in sufficient 174 quantities, confer a health benefit. This is something that needs to be mentioned in your introduction, does not really fit here.
Line 182-203: Put this in a table if you just sum up or reform this that it is nice to read.
Line 211: but can occur at any site of the body.
Line 212: High prevalence in Western countries!
Table 2: Look at the structure of the table. Different fonts used..
Line 418- 442: Work with a table or make a nice coherent text on this, do not use summation signs.
Include reference at the end of the sentence, also include references in your tables.
Line 515: 2.15. The gut microbiota in patients with rosacea, also reports of SIBO in rosacea patients!
Line 536: reform sentence
Overall, a lot of repetition from an earlier seen review. Please do not repeat, but focus on the gut microbiome in these specific skin pathologies and potential oral and cutaneous tested probiotics? Further, genera are not italicized, references are missing and structure of several sentences needs to have a second look at. Tables do not have a good subscript associated with them, lack a nice structure and do not have references with them.
Author Response
Reviewer 2 Q@A:
Major comments:
Reading the abstract suggests already that this review is not renewing from the review “Gut-skin axis: current knowledge of the interrelationship between microbial dysbiosis and skin conditions” that was earlier seen in MICROORGANISMS 9(2). Repetition of four diseases that were also described in there, new information is lacking.
The article here included aims to review the main dermatological diseases, their relationship with the human microbiota, but specially the effect of probiotics usage in the three skin diseases where there is more evidence on the use of probiotics.
See reference: De Pessermier 2021 Gut–Skin Axis: Current Knowledge of the Interrelationship between Microbial Dysbiosis and Skin Conditions https://www.mdpi.com/2076-2607/9/2/353
And if not mistaken, this paper is not referenced.
It is referenced now. Reference [2]
Line 92: 1.2. The intestinal microbiota and its involvement in dermatological processes: Only described the impact of the gut microbiome on the skin through the endocrine systems. However, the gut-skin axis is the result of the complex interplay between nervous system, immune system, endocrine systems, and environmental factors, which is worth to mention I believe. In addition, they react with one another in a bidirectional manner.
In this new version of the article, sited in the introduction of each of the sections dedicated to atopic dermatitis, psoriasis and acne we describe more details about the mechanisms implicated in this diseases and the relacion with the gut skin axis, but the focus remains on the use of probiotics for these pathologies
Minor comments:
Genus taxa are not italicized
Bacterial species and genus names have been corrected and written in Italian.
Line 31-31 the therapeutic role of the probiotic has been explored include atopic dermatitis with both intervention and primary prevention studies, psoriasis, acne, and rosacea.
Now line 27. The word “Include” has been changed for in.
The probiotic suggests that there is only one type of probiotic available with therapeutic effects? Suggestion: the therapeutic role of probiotics … Also, the structure of the sentence needs to be reformed.
It has been changed in the new version of the manuscript.
Line 70: secretory IgA, sIgA instead of IgA S, IgA, no space between Ig and A
Now line 64. It has been changed.
Line 83: The SCFAs instead of The SCFA
Now line 77. It has been changed.
Line 99: Reform sentence: it is at between 2 and 10% in adults, ranging from one territory to another, and from 15 to 30% in children
Now line 95. It has been changed.
Line 100-101: Add references: It is strongly 100 associated and can coexist with other allergic, immunological or food intolerances.
Now line 97. Reference [15] has been added.
Line 134-141: Missing references
Now line 130-137. References [23,24] has been added.
Table 1: Failure os the above
Previous Table 1 has been deleted.
Line 146: pH
Now line 140. It has been amended.
Line 174: Probiotics are defined as those microorganisms that, when administered in sufficient 174 quantities, confer a health benefit. This is something that needs to be mentioned in your introduction, does not really fit here.
It has been changed to Introduction, line 25.
Line 182-203: Put this in a table if you just sum up or reform this that it is nice to read.
Summary table has been added. Line 206.
Line 211: but can occur at any site of the body.
Now line 213. It has been added.
Line 212: High prevalence in Western countries!
Now line 214-215. It has been added.
Table 2: Look at the structure of the table. Different fonts used.
Previous Table 2 has been deleted.
Line 418- 442: Work with a table or make a nice coherent text on this, do not use summation signs.
Summary table has been added. Line 445.
Include reference at the end of the sentence, also include references in your tables.
It has been changed.
Line 515: 2.15. The gut microbiota in patients with rosacea, also reports of SIBO in rosacea patients!
The chapter dedicated to rosacea has been eliminated. The article is now focused in the three skin diseases with more evidence on the use of probiotics as coadjuvant treatment.
Line 536: reform sentence
The chapter dedicated to rosacea has been eliminated.
Overall, a lot of repetition from an earlier seen review. Please do not repeat, but focus on the gut microbiome in these specific skin pathologies and potential oral and cutaneous tested probiotics? Further, genera are not italicized, references are missing and structure of several sentences needs to have a second look at. Tables do not have a good subscript associated with them, lack a nice structure and do not have references with them.
Bacterial species and genus names have been corrected and written in Italian.
The treatment tables have been deleted and summary tables have been added.
Reviewer 3 Report
the topic are very interesting and the organisation of the paper are simple enough to never lost the reader.
1/ however for atopic dermatitis and probiotics some main paper on this field are missing that The WAO guideline panel suggests using probiotics in pregnant women at high risk for allergy in their children, and in infants at high risk of developing allergies, because there is a net benefit resulting primarily from prevention of eczema. So please take a look back to the litterature to be up to date and rely on the validated probiotic usage better to expose some potential benefit in some stage on allergies.
2/ for acne please review since you are very oftenly speak about cutinobacterium in place of cutibacterium acnes.
3/ for rosacea no real element lead to over promising paragraph with probiotics
Author Response
Reviewer 3
1/ however for atopic dermatitis and probiotics some main paper on this field are missing that The WAO guideline panel suggests using probiotics in pregnant women at high risk for allergy in their children, and in infants at high risk of developing allergies, because there is a net benefit resulting primarily from prevention of eczema. So please take a look back to the litterature to be up to date and rely on the validated probiotic usage better to expose some potential benefit in some stage on allergies.
We have included new bibliography in this version of the paper.
2/ for acne please review since you are very oftenly speak about cutinobacterium in place of cutibacterium acnes.
It has been corrected.
3/ for rosacea no real element lead to over promising paragraph with probiotics
The chapter dedicated to rosacea has been eliminated.
Reviewer 4 Report
The authors performed a review on the data considering intestinal microbiota in relation to skin diseases. I have two major concerns that should be addressed by the authors in the revised manuscript:
- According to current standards, such review should be amended with some methodology part, including e.g. search terms, databases searched, number of papers identified, number of papers qualified for analysis, etc.
- The current manuscript contains four tables. However, they are very general and irrelevant to the review aim. Please delete them. However, please prepare tables summarising your major findings or describing shortly major studies included in the analysis.
Author Response
Reviewer 4
The authors performed a review on the data considering intestinal microbiota in relation to skin diseases. I have two major concerns that should be addressed by the authors in the revised manuscript:
- According to current standards, such review should be amended with some methodology part, including e.g. search terms, databases searched, number of papers identified, number of papers qualified for analysis, etc.
Based in the experience of the authors in the use of probiotics in some skin diseases and the previous publications during last 5 years, a summary of recent findings related with the use of probiotics in these dermatological diseases is given in this article.
In this case authors do not pretend to publish a meta analysis with a methodology related to the selection of the articles or the clinical trials evaluated and finally included in the reference section of this article. This is way we consider that the methodology selecting articles is not necessary to be included in the introduction nor add a methodology section
with a brief report of this unnecessary information. One example of this argument that appears in other articles is the article by Possarem et al, that I personally reviewed in the peer review process months ago. In this article a brough description of different skin diseases that authors hypothesize could be related with the microbiome is included, although most of these diseases probably will never be related with dysbiosis and specially won’t be related with probiotics treatment. This is way the article here attached includes only the more relevant skin diseases that at nowadays have been related with indications about the use of probiotics as coadjutant treatment.
- The current manuscript contains four tables. However, they are very general and irrelevant to the review aim. Please delete them. However, please prepare tables summarising your major findings or describing shortly major studies included in the analysis.
The treatment tables have been deleted and other summary tables have been added.
Round 2
Reviewer 4 Report
The authors have significantly improved their review. However, I disagree, that they should not provide any methodological issues. For E.g. which databases were reviewed? How they select papers to be included in the review? These simple pieces of information have nothing to do with metaanalysis. They just give the readers some hints, how reliable the review is.
Author Response
The point by point, details of the revision to the manuscript and our responses to the reviewer
- Question: According to current standards, such review should be amended with some methodology part, including e.g. search terms, databases searched, number of papers identified, number of papers qualified for analysis, etc.
Response: Authors include now in the introduction section a brief paragraph with the explanation about this matter. Please find the information remarked in yellow in the new version of the manuscript.